# Evaluation of high resolution snowpack simulations from global datasets and comparison with Sentinel-1 snow depth retrievals in the Sierra Nevada, USA

Laura Sourp[1,2], Simon Gascoin[1], Lionel Jarlan[1], Vanessa Pedinotti[2], Kat J. Bormann[3], Mohammed Wassim Baba[4]

[1]Centre d'Etudes Spatiales de la Biosphère, CESBIO, CNES/CNRS/INRAE/IRD/Université Toulouse 3 Paul Sabatier, 31401 Toulouse, France
[2]MAGELLIUM, Ramonville Saint-Agne, 31520, France
[3]Airborne Snow Observatories, Inc., Mammoth Lakes, CA, United States
[4]Science, Applications & Climate Department, European Space Agency, Frascati 00044, Italy

*Correspondence to*: L. Sourp (laurasourp@gmail.com)

**Abstract.** Spatial distribution of mountain snow water equivalent (SWE) is key information for water management. We implement a tool to simulate snowpack properties at high resolution (100 m) by using only global datasets of meteorology, land cover and elevation. The meteorological data are obtained from ERA5 which makes the method applicable in near real time (5 day latency). We evaluate the output using 49 SWE maps derived from airborne lidar surveys in the Sierra Nevada. We find a very good agreement at the catchment scale using uncalibrated lapse rates. Larger biases at the model grid scale are especially evident at high elevation but do not alter the catchment-scale snow mass accuracy. We additionally compare the simulated snow depth to Sentinel-1 retrievals and find a similar accuracy with respect to synchronous airborne lidar surveys. However, Sentinel-1 snow depth products are sparse and often masked during the melt season, whereas ERA5-SnowModel provides spatially and temporally continuous SWE.

## 1 Introduction

Many populated regions with dry summers and wet winters depend on mountain snow for water supply (Mankin et al., 2015; Sturm et al., 2017; Viviroli et al., 2020). Understanding the catchment scale seasonal snow storage before and during the melt season is key to optimizing water use between hydropower production, crop irrigation and freshwater supply. In addition, an accurate prediction of the timing and magnitude of the snowmelt runoff is bound by our ability to characterize the spatial distribution of mountain snow before the melt season (Freudiger et al., 2017).

Despite its hydrological significance, the snow water equivalent (SWE) remains poorly monitored in many mountain regions especially outside North America and Europe. In situ measurements are often too sparse considering the spatial variability of mountain snow (Fayad et al., 2017). To cope with this issue, airborne measurement campaigns are now routinely used in the western USA to measure snow depth but their cost remains prohibitive in other regions (Painter et al., 2016). Meanwhile, several approaches have emerged to retrieve mountain snow depth from satellite remote sensing (e.g. Pléiades, ICESat-2 and Sentinel-1). Pléiades very high resolution stereoscopic images can be used to generate snow depth images by differencing two digital elevation models. However, this approach is limited to small regions (Marti et al., 2016)    ICEsat-2 lidar altimetry has the potential to provide    snow depth data at global scale but with    a sparse sampling (Deschamps-Berger et al., 2023). Sentinel-1 has been used to derive snow depth at 1 km resolution in the northern hemisphere (Lievens et al., 2019), and 500 m over the European Alps (Lievens et al., 2022). This method, which is based on an empirical change detection method applied to the cross-polarization ratio, is limited to dry snow conditions and therefore does not allow monitoring of the snowpack during the melt season. However, it offers a global and spatially continuous coverage which is a key advantage with respect to the other approaches. All the above remote sensing approaches require an estimation of snow density to obtain the SWE, but it has been established that snow depth explains most of the SWE variance (Guyennon et al., 2019; López-Moreno et al., 2013; Sturm et al., 2010; Bormann et al., 2013).

Another approach to estimating mountain SWE distribution is to use a snowpack model, but the challenge then lies with obtaining accurate meteorological forcing (Günther et al., 2019; Raleigh et al., 2016). To cope with the lack or sparsity of in situ meteorological measurements, one solution is to use atmospheric model outputs as forcing data. In particular, climate reanalyses can provide long term hourly meteorological data at global scale. Climate reanalyses are also becoming increasingly accurate (Hersbach et al., 2020) with advances in atmospheric and land surface modeling and the assimilation of a growing dataset of in situ and remote sensing observations. These reanalyses have also seen notable progress in recent years in terms of latency. For example, the preliminary ERA5 reanalysis provided by the European Centre for Medium-Range Weather Forecasts has a short latency of 5 days (whereas it was 2–3 months with the previous ERA-Interim). This preliminary product only rarely deviates from the fully quality-checked final product that is released 2 months later (Hersbach et al., 2020). This timely product can fulfill the need for up-to-date meteorological forcing information. However, reanalyses cannot be used directly to force a mountain snowpack model because the grid cell size is too coarse (approximately 30 - 50 kilometers for ERA5 and MERRA-2 respectively), which creates large biases in the computed SWE (Wrzesien et al., 2019; Liu et al., 2022).

To address the mismatch in spatial resolution between reanalyses datasets and snow distribution, previous studies used downscaling algorithms based on a digital elevation model before running a snowpack model on a finer grid (Armstrong et al., 2018; Baba et al., 2018; Billecocq et al., 2023; Mernild et al., 2017; Weber et al., 2021). This approach enables estimation of high resolution SWE and snow depth without ground data. For example, Mernild et al. (2017) and Baba et al. (2018) studied the snowpack properties over large and ungauged regions in the Andes and the High Atlas mountain ranges using the

MicroMet/SnowModel package (Liston et al., 2020; Liston and Elder, 2006a, b). The evaluation of these simulations relied on in situ observations or remote sensing snow cover area. Weber et al. (2021) used 10 years of snow depth measurements from two automatic weather stations to assess their simulations in the Research Catchment Zugspitze (12 km²). Mernild et al. (2017) used 13 years of MODIS data over the Andes Cordillera (~16 million km²) along with 4 km grid maps of snow depth that were reconstructed from in situ observations. Baba et al. (2018) used 18 years of MODIS data to assess simulations in the High Atlas of Morocco, snow depth at a single automatic weather station, precipitation at three meteorological stations and river discharge of the Ourika catchment (503 km²). However, in situ data are sparse and MODIS snow cover area does not allow a thorough evaluation of the model ability to capture snow mass across the landscape.

In this study, we focus on the Tuolumne River catchment in the Sierra Nevada, USA (Figure 1). Since 2013, this site has been regularly surveyed by the Airborne Snow Observatory (ASO) to determine snow depth and SWE. The ASO dataset on the Tuolumne catchment     is the densest time series of high resolution snow depth (3 m) and SWE (50m) maps publicly available at this scale (1100 km²) in the world. The dataset contains 49 surveys and spans several years with contrasted climatic conditions including California's most severe drought in the last 1200 years during 2012-2014 (Griffin & Anchukaitis, 2014) and the "snowpocalypse" 2016–2017 winter which was characterized by near-record snow accumulation (Painter et al., 2017). We leverage this observational dataset to evaluate a new processing pipeline which generates 100 m resolution SWE and snow depth estimates from ERA5 or ERA5-Land. This pipeline, inspired by previous works (Baba et al., 2018; Mernild et al., 2017) is a wrapper around MicroMet/SnowModel code. It was designed to work with global meteorological forcing datasets. As such, the workflow can generate high resolution snow cover simulations in any region of interest across the globe from 1940 up to present, with any resolution between 1 m and 200 m (Liston and Elder, 2006b). Furthermore, we compare the output of this pipeline with the more direct approach of Sentinel-1 snow depth on dates matching the ASO measurements.

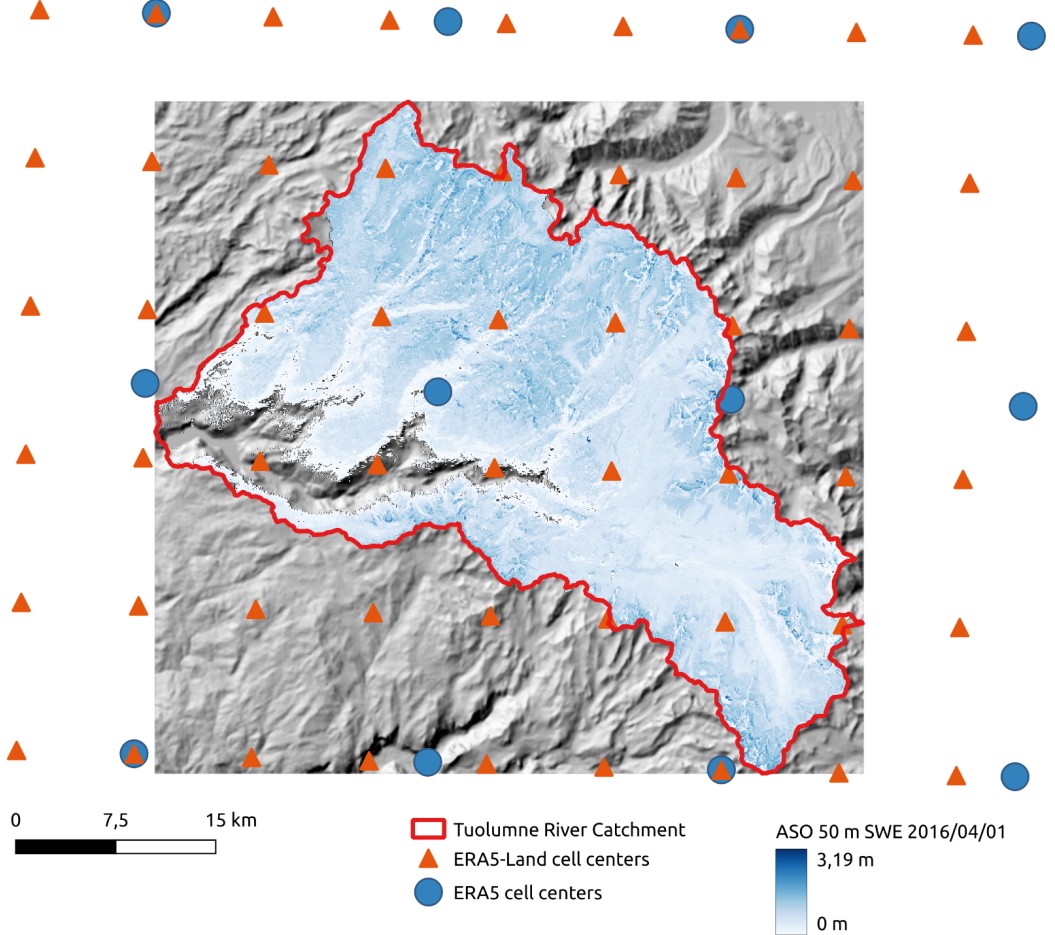

**Figure 1: Map representing the SWE variability measured by ASO, along with ERA5 and ERA5-Land cells centers and the Tuolumne River catchment border overlaying the DEM hillshade.**

## 2 Data and Methods

### 2.1 Data

We used two reanalyses in this study, ERA5 and ERA5-Land. ERA5 is a reanalysis of the global climate and weather since 1940, with a 0.25° resolution (approximately 30 km). It provides hourly atmospheric, oceanic and land-surface variables computed with a global model and improved by the assimilation of multiple in situ and remote sensing datasets (Hersbach et al., 2020). ERA5-Land is produced by recomputing ERA5 land variables at finer resolution using a downscaled meteorological forcing (Muñoz Sabater, 2019). It delivers these variables on a global scale at a 0.1° resolution, from 1950 to this day. As mentioned above, preliminary versions of ERA5 and ERA5-Land are distributed with a short latency of 5 days. These datasets

are freely available from the Copernicus Climate Change Service (C3S) and can be queried via their application programming
interface (with tutorials that can be found on their website : Retrieving data — Climate Data Store Toolbox 1.1.5
documentation)
. We focused on ERA5 here as we found that it yielded slightly better results than MERRA-2 in a previous case study using
the same approach (Baba et al., 2021). In addition, the latency of MERRA-2 is 3 weeks which may be too long for operational
water resources applications. To run the model (see section 2.2.1    ), we also used the 30 m Copernicus Digital Elevation
Model (DEM) (Copernicus Digital Elevation Model, 2023) and the 100 m Copernicus Land Cover (Buchhorn et al., 2020).

We obtained Sentinel-1 snow depth between 2016 and 2019 from the C-SNOW repository (C-SNOW). Sentinel-1 C-band
backscatter observations were used to derive ~1 km resolution snow depth, using an empirical change detection (Lievens et
al., 2019). This product has a revisit time of approximately 3 days over the Tuolumne River catchment during winter but
provides almost no data in spring because the algorithm is considered to be invalid when the snowpack contains liquid water.
When the snowpack is wet, there is a larger absorption and reflection of the microwave signal emitted by Sentinel-1 which
greatly decreases the performances of the C-SNOW algorithm (Lievens et al., 2019; Tsai et al., 2019).

For the evaluation of model outputs and Sentinel-1 products, we used 49 SWE and snow depth maps collected between 2013
and 2019 by the ASO. The ASO acquires hyperspectral data for snow albedo and lidar data for snow depth and computes SWE
as a derived product (Painter et al., 2016). Snow depth is available with a 3 m resolution while SWE has 50 m resolution. The
reported accuracy on the 3 m snow depth products is 0.08 m (Painter et al., 2016) and from spatially intensive sampling, the
reported accuracy for the 50 m snow depth products is < 0.01 m (Painter et al., 2016, Figure 15). There are no published
references for the 50 m SWE product. However, Rayleigh & Small (2017) estimated an uncertainty in modeled density of 48
$kg/m^3$ in the Tuolumne basin. This uncertainty can be regarded as a conservative estimate as in situ measurements of snow
density are also used by the ASO to adjust their density model (Painter et al., 2016). Therefore, for a 1 m deep snowpack and
an    uncertainty in snow density of    $50 kg/m^3$    , we estimate the uncertainty of the 50 m SWE products to be    0.05 m
w.e (meters of water equivalent)    .
**2.2 Methods**
**2.2.1 SnowModel**
SnowModel is designed to simulate snow evolution on a high resolution grid (1 m to 200 m increments) and a time step from
1 min to 1 day (Liston et al., 2020; Liston and Elder, 2006a). It is separated into four submodels: i) MicroMet redistributes
meteorological forcings (air temperature, relative humidity, wind speed and direction, precipitation, solar radiation, long wave
radiation, and surface pressure) to the target simulation grid (Liston and Elder, 2006b). ii) EnBal computes the snow surface
energy balance, iii) SnowPack computes the snow density and snow depth and iv) SnowTran-3D computes the blowing snow
sublimation and snow redistribution due to wind transport (Liston et al., 2007). SnowModel accounts for the vegetation effects
on the snow cover such as coniferous forests or grassland to the grid cell vegetation type. MicroMet was originally designed
to interpolate station data on a regular grid. Here, a climate reanalysis grid cell is considered as a virtual station located at the
grid cell center.

**2.2.2 Model input**

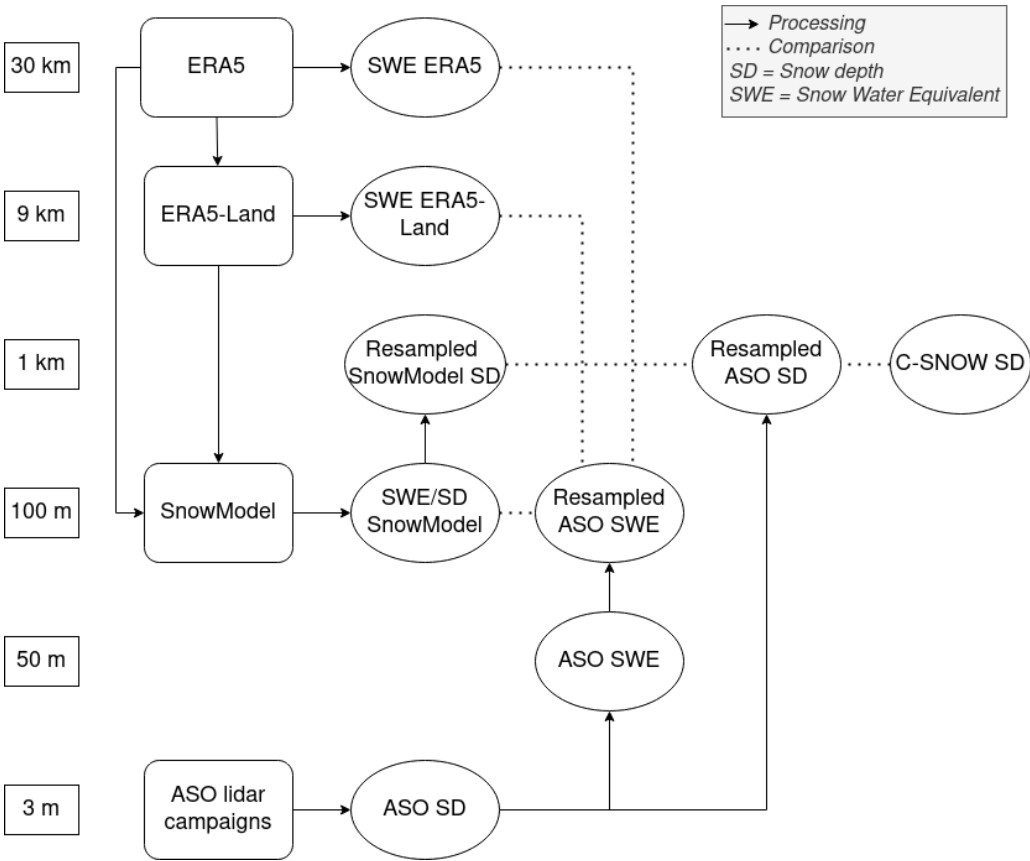

**Figure 2: Summary of the different data sources, with their spatial resolutions. Arrows represent a process and the**
**dotted lines the comparison between different data.**

We developed a tool to automatically prepare SnowModel input files from ERA5 and ERA5-Land data and run the simulations.
This tool uses a DEM of the region of interest as an input along with the start and end of the simulation period. We let the
user specify the DEM because it is used to define the model grid, which is the main control of the computation time. Here we
used the 30 m Copernicus orthometric DEM that we extracted and resampled to a WGS84 UTM 11N grid at 100 m resolution
using the bilinear method over a region covering the Tuolumne River catchment. The simulation period was set to September

2012-August 2019, and spans seven years of snowpack dynamics. Using the Climate Data Store Application Program Interface, our tool downloads ERA5 or ERA5-Land hourly meteorological data ( 2 m temperature, 2 m dew point temperature, precipitation, 10 m wind eastward and northward component) over the region of interest given by the DEM bounding box extended to the adjacent ERA5/ERA5-Land neighbouring cells (~30km/11km respectively). Once downloaded, the meteorological data are processed to match SnowModel/MicroMet input format and

units. ERA5-Land precipitation is provided as daily cumulative values and is therefore converted to hourly precipitation rate. Wind components (u,v) are converted into wind speed and direction (0-360°N). The dew point temperature is converted into relative humidity using Buck's equation (Buck, 1981), the same equation that is used in MicroMet. The elevations of ERA5/ERA5-Land cells are determined from the global geopotential file that is first interpolated on the model grid with a bilinear algorithm. The tool also resamples the Copernicus land cover map on the model grid using the mode resampling algorithm (GDAL/OGR contributors, 2024). We built a correspondence table to remap the Copernicus land cover classes to the SnowModel land cover classification (see Table A1 in appendix). We set all SnowModel parameters (the curvature length scale, curvature and wind slope weights, minimum wind speed, precipitations schemes for downscaling or for rain-snow fractions, subcanopy radiations schemes, various thresholds for wind transport calculations) to the default values (see the parameter file snowmodel.par in the code availability section) . A simple parametrization of the albedo is used with a constant value 0.8 in dry condition, whereas albedo values for melting snow cover are set according to land covers (Liston et al., 2020). We used the default monthly temperature lapse rates and precipitation factors which adjust the precipitation values to the elevation of the model grid. This tool is implemented in Python. The source code and a more detailed documentation is available at (code availability section).

### 2.2.3 Comparison with ASO SWE

We resampled the ASO SWE (n=49 surveys) to the model grid which has a resolution (100 m). The resampling was done using the weighted average of all valid contributing pixels (GDAL/OGR contributors, 2024). We also created a validity mask to select cells in the Tuolumne River catchment that were always observed by the ASO during this period (some regions were not always available, representing 2.5% of the catchment area). ASO data and ERA-SnowModel outputs were averaged over the valid cells to compute the temporal evolution of the catchment-mean SWE. Then, we analyzed the spatially distributed residuals on the catchment for each observation date of a dry year (2014-2015), a wet year (2016-2017) and an average year (2015-2016). We used the validity-masked SWE maps to subtract the ASO observations from the ERA-SnowModel output. A positive bias means the simulated SWE is larger than the observations.

Additionally, we extracted ERA5 and ERA5-Land daily SWE over the Tuolumne River catchment and computed the catchment scale SWE using an area weighted average (i.e. each SWE value was weighted by the fraction of the grid cell area within the catchment). Since these SWE products have a very coarse resolution of approximately 31 and 9 km ( Fig. 1, Fig. 2), we did not use them to analyze the residuals distribution as above.

**2.2.4 Comparison with Sentinel-1 snow depth**

Over the entire study period, we identified three matchup dates for which we have both ASO and Sentinel-1 snow depth observations with a minimum coverage of 60% of the catchment area. On these dates, the snow depths given by ASO, Sentinel-1 and ERA-SnowModel were resampled to a common 1 km UTM grid. We applied another validity mask for the cells where the snow depth is not always available to all three snow depth datasets (here representing 8.5% of missing data in the catchment). The missing values in the 3 m resolution ASO dataset are propagated at the 1 km resolution validity mask. This decreases the number of observations but ensures that the resampled 1 km snow depths maps are not biased by the spatial distribution of non-valid pixels in  the 3 m ASO snow depth dataset.     We computed the distributed residuals by subtracting the ASO snow depth from both SnowModel simulations and Sentinel-1 data. For each date, we averaged the residuals to compute the mean bias, and we computed the standard deviation of the error. We also computed the RMSE over the catchment for each date .

**3 Results**

**3.1 Comparison with ASO SWE**

Figure 3 shows the temporal evolution of the catchment scale SWE from ASO observations and SnowModel simulations forced with ERA5 and ERA5-Land. There is a very good agreement between the observations and both simulations, with an overall correlation of 0.99 for both ERA5 and ERA5-Land SnowModel simulations (with 49 observation dates). First, both simulations capture the large interannual variability of SWE in the Tuolumne River catchment during the study period. The observed annual peak SWE ranges from 0.11 m in 2015 to 1.27 m in 2017 while the SnowModel simulations yield from 0.17 m to 1.19 m with ERA5 and from 0.12 m to 1.24 m with ERA5-Land during the same years (but at different dates). In addition, the model is reproducing the seasonal evolution of SWE with an annual RMSE ranging from 0.03 m to 0.13 m. The catchment scale SWE accumulation in the ERA5-SnowModel simulations is well captured.We note an underestimation of the snow ablation rates in late spring, which causing a delay from a few days (2013) to one month approximately (2019) in the date of complete melt out. This issue is mostly evident in 2016-2017 since the ablation rates are insufficient to reach the complete removal of the snowpack in August as observed by the ASO. Interestingly, we also note that ERA5-Land without resampling almost always reports the lowest RMSE at the catchment scale, though at 0.1º the distribution of the snow is not well represented.

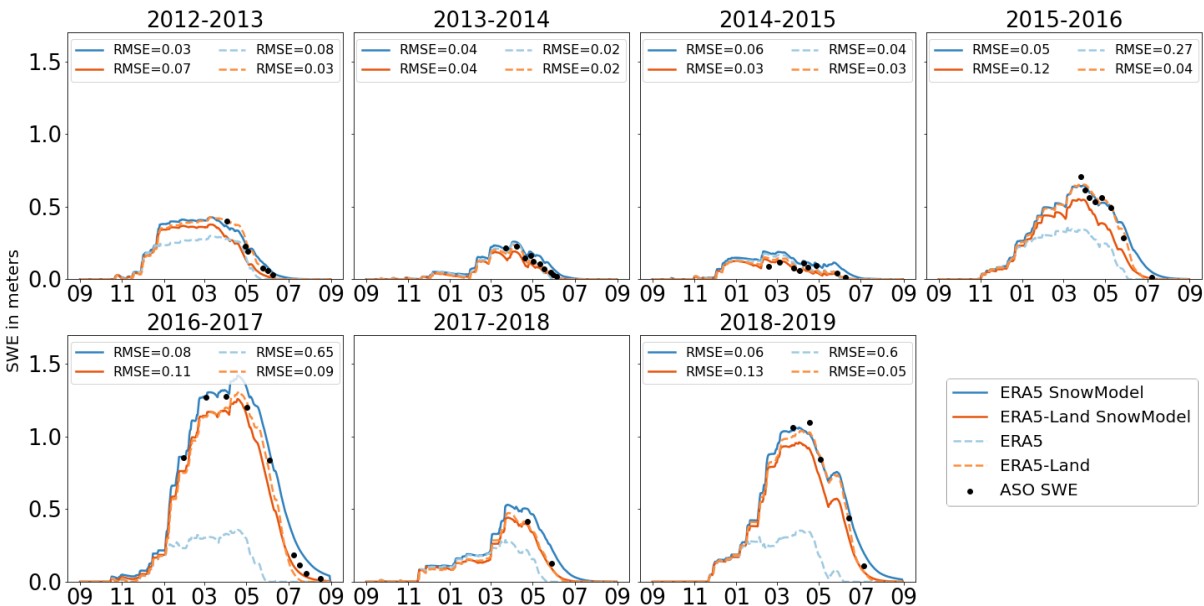

**Figure 3: Temporal evolution of the Tuolumne river catchment SWE for seven hydrological years from 2012 to 2019. The legend indicates the RMSE between the simulated SWE and the ASO SWE for each year.**

To go beyond this coarse catchment scale diagnostic (1100 km²), we also analyze the distribution of the residuals at the pixel scale (0.01 km²). We computed a map of RMSE using all the 49 validation dates we have between 2013 and 2019. 10% of the cells in this map have a RMSE above 0.5 m w.e.. Figure 4 shows the distribution of the residuals for every date with ASO observations for three contrasted hydrological years. The spread of the residuals are shown with the interquartile (i.e., the difference between the 25 and 75th percentiles) inside the colored boxes, and with the 5-95th percentiles inside the whiskers. This figure indicates that the spread of the residuals increases with the mean SWE depth. For the dry year, the interquartiles of SnowModel SWE residuals for ERA5 and ERA5-Land do not exceed 0.17 m and 0.09 m w.e. respectively. For the average year, the interquartiles reach 0.31 m and 0.38 m w.e. and for the wet year 2017, they peak respectively at 0.64 and 0.82 m w.e.

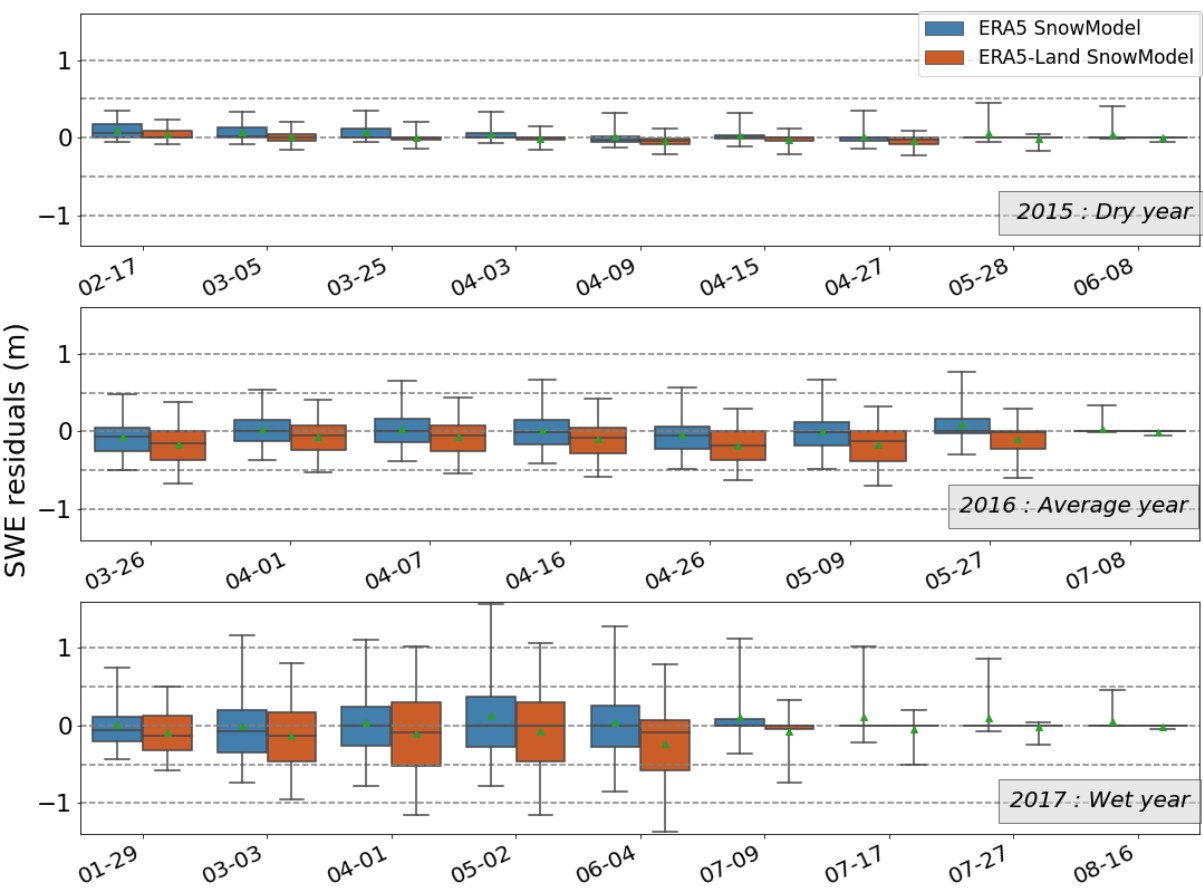

**Figure 4: Distribution of the residuals between the SnowModel simulated SWE and the ASO SWE at 100 m resolution in the Tuolumne river catchment (in m w.e.) for three contrasted hydrological years. Filled boxes represent the interquartile range, the whiskers show the 5-95 percentiles, the line in each box represents the median of the distribution, and the green triangle shows the mean.**

Figure 5 shows the distribution of the residuals for two dates (2016-04-01 and 2016-05-27) by slope, elevation and aspect. We aimed to distinguish the model performance in terms of accumulation and ablation processes to better separate the sources of uncertainties in future studies. Therefore we selected a date before the melting season (April 01 2016) and a date near the end of the melting season (May 27 2016)     . The interquartile of the error distribution never exceeds 0.41 m.w.e. in slope or aspect categories but peaks at 0.67 m.w.e. in the highest elevation band the 1st of April for the simulations forced with ERA5-Land.

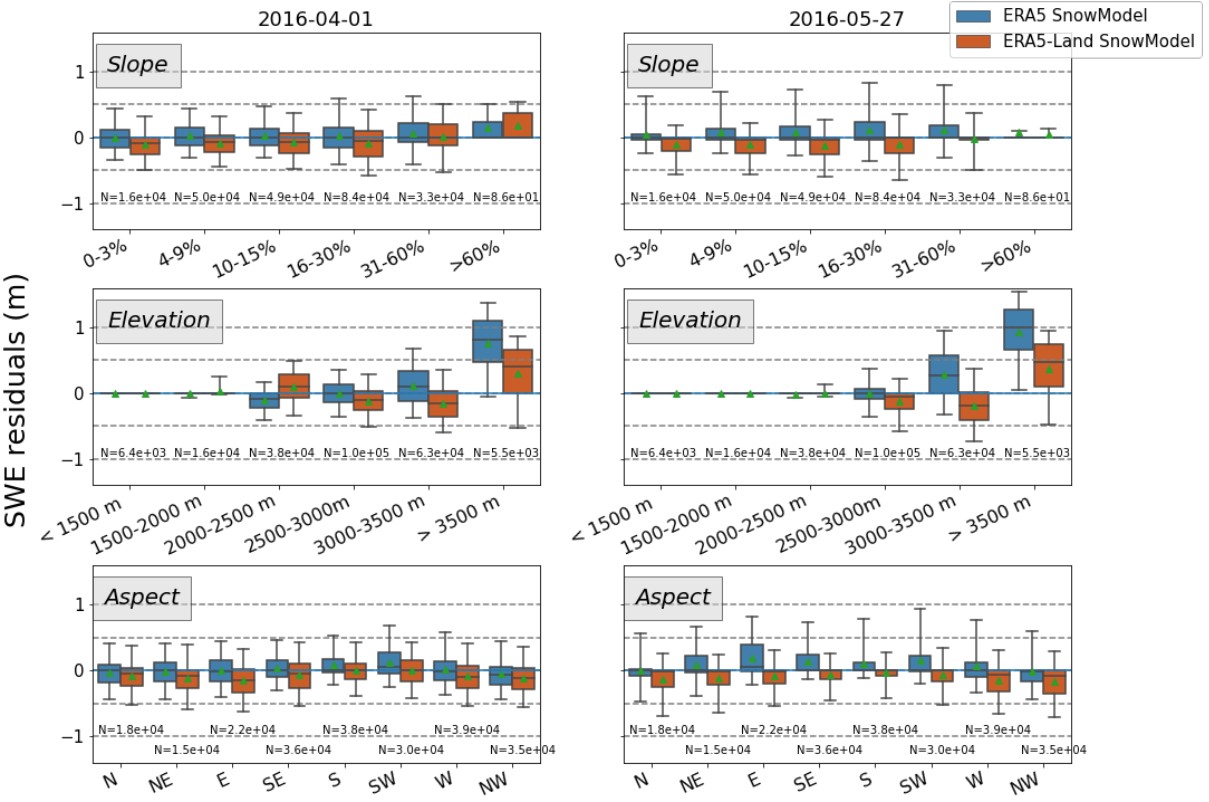

**Figure 5: Distribution of the residuals between the SnowModel simulated SWE and the ASO SWE at 100 m resolution in the Tuolumne river catchment (in m w.e.) on the 1st of April 2016 (left) and the 27th of May (right), stratified by slope (in percent), elevation (in m a.s.l.) and aspect (in degrees from north). Whiskers show the 5-95 percentile, the line in each box represents the median of the distribution and the green triangle shows the mean. Slope, elevation and aspects have been calculated using the DEM at 100 m resolution.**

## 3.2 Comparison with Sentinel-1 snow depth

Between 2016 and 2019, there are three dates for which we have both Sentinel-1 and ASO snow depth data. Figure 6 presents snow depth maps on the Tuolumne River catchment at 1 km resolution with Sentinel-1, ASO and ERA5-SnowModel data. Some pixels are not always observed with ASO data and these missing values are propagated at 1 km resolution (if there is at least one missing value among the contributing pixels, a missing value is attributed to the target 1 km cell). The same mask is applied on the SnowModel simulations and Sentinel-1 data. Additional missing values are observed in the Sentinel-1 snow depth maps. Therefore, the statistics of Figure 7 are not computed on the exact same area. We chose to take all possible data into account.

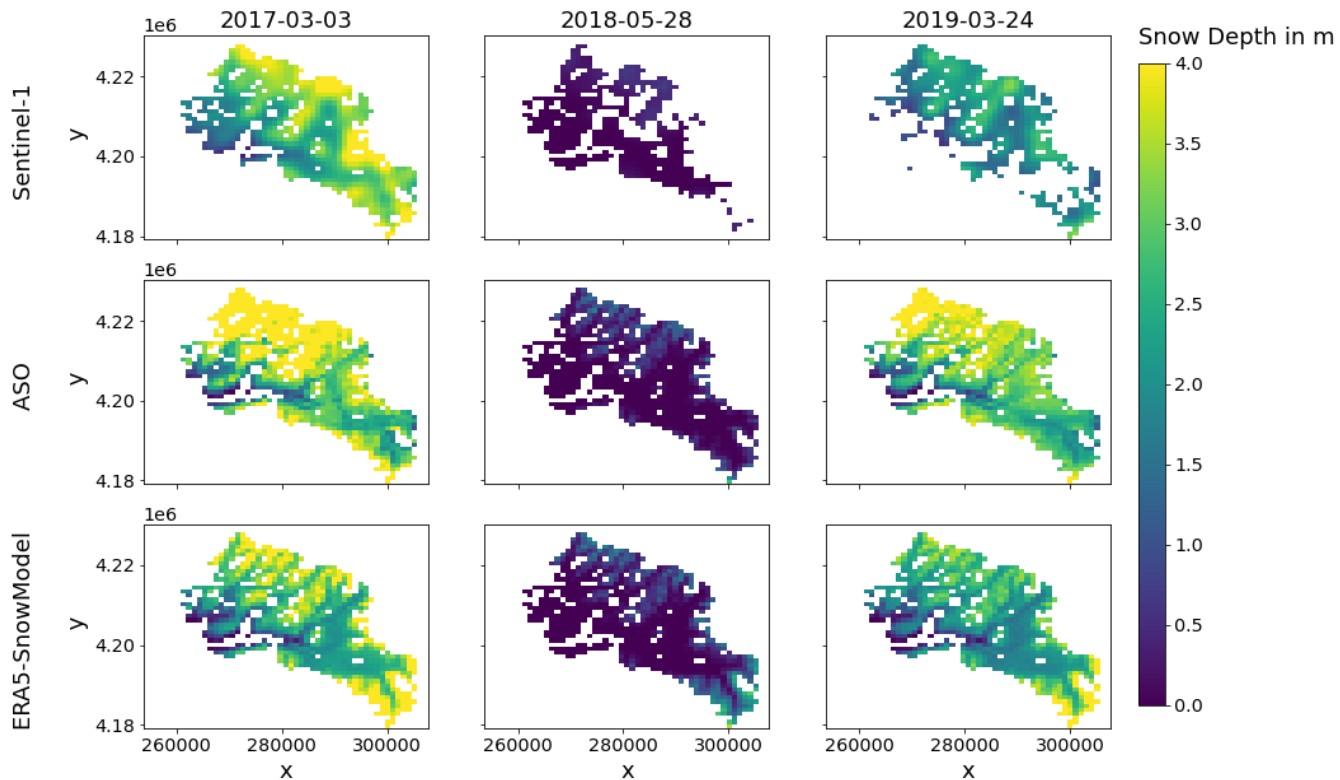

**Figure 6: Snow depth maps at 1 km resolution with Sentinel-1, ASO and ERA-SnowModel data.**

Figure 7 shows the Sentinel-1 observed and SnowModel simulated snow depth compared to the ASO observed snow depth, resampled to a 1 km resolution. On the 2017-03-03, Sentinel-1 has the lower bias (-0.43 m), standard deviation (0.86 m) and RMSE (0.96 m). These statistics are close    to the ERA5-SnowModel simulations (respectively -0.49 m, 0.9 m, 1.02 m) while ERA5-Land-SnowModel simulations have a greater bias (-0.83 m) and RMSE (1.2 m) with a comparable standard deviation (0.86 m). On the second date, the 2018-05-01, Sentinel-1 still performs the best with a bias of -0.05 m, and standard deviation and RMSE both equals to 0.21 m    . On this date,    ERA5-Land-SnowModel simulations are similar to Sentinel-1 with a bias of -0.09 m, standard deviation of 0.26 m and RMSE of 0.27 m; while ERA5-SnowModel simulations underperform with a 0.16 m bias, a 0.41 m standard deviation and a 0.44 m RMSE.. Finally on the 2019-03-24, the closer data to the ASO snow depths seems to be the ERA5-SnowModel simulations with an bias of -0.65 m, a standard deviation of 0.81 m and an RMSE of 1.04 m.       Sentinel-1 data have the highest bias (-1.24 m) and RMSE (1.38 m), but the lowest standard deviation (0.61 m)    . ERA5-Land-SnowModel simulations also have a high bias (-0.92 m) and RMSE (1.17 m), with a standard deviation of 0.73 m.  We see an underestimation of the snow depth above 2 meters with Sentinel-1 in 2017 and 2019, which is very clear for 2019 when the mean bias is the highest with a relatively low standard deviation. In 2018, both the ASO and Sentinel-1 observed really low snow depths (<1 m) but there is still a negative bias (-0.05 m) in the Sentinel snow depth distribution    .

260  With the ERA5 SnowModel simulations, most of the distribution is centered around a     negative bias that is underestimating
261  the snow depth in 2017 and 2019. We note several cells with a high positive error. In 2018, the situation is reversed : most of
262  the snow depth estimated with ERA5 SnowModel are overestimated. Finally, the simulations with ERA5-Land seem to cap at
263  4 meters of snow depth in 2017 and 2019, with a declining accuracy with the ASO snow depth starting at 2 m. In 2018, the
264  ERA5-Land SnowModel simulations are mostly underestimating snow depths.

265

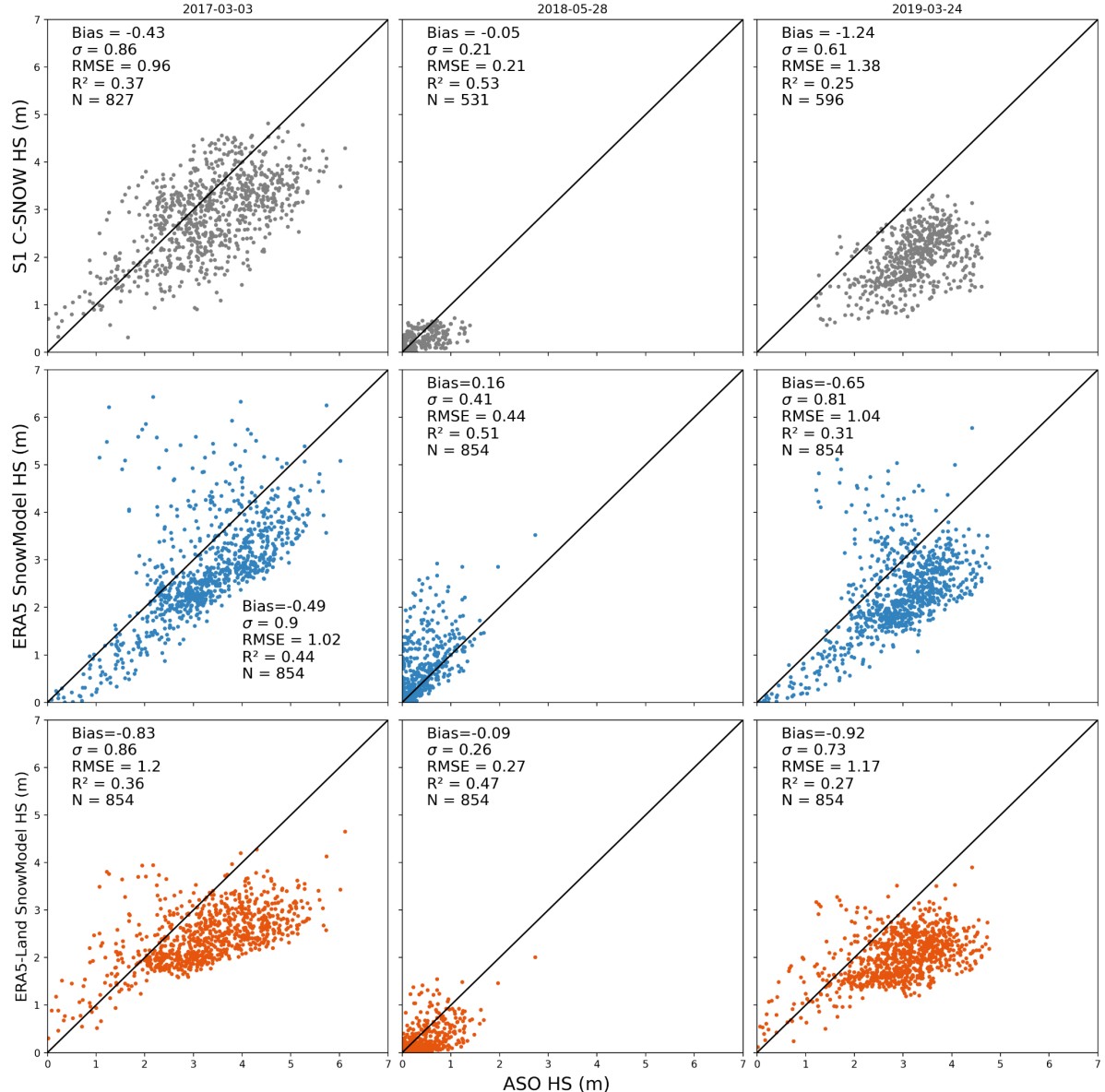

266

**Figure 7: Scatter plots representing the observed and SnowModel simulated snow depth data as a function of ASO snow depth data, with a one to one line in black. All data are resampled at 1 km resolution. N is the number of values in each plot.**

## 4 Discussion

Downscaling ERA5 forcing is critical to obtain realistic SWE in the Tuolumne catchment and is sufficient to remove the strong negative bias that is otherwise present in the original ERA5 SWE (Fig 3). The use of this pipeline for long simulation periods could also bypass the discontinuities in the ERA5 SWE (Urraca and Gobron, 2023) which are caused by a snow capping in the data assimilation code and the arrival of new snow depth data available for assimilation. The main effect of the downscaling is a better representation of the air temperature distribution and therefore a better representation of the solid precipitation fraction. Then, the performance of the SnowModel simulated SWE largely relies on ERA5 precipitation. Our results suggest that the winter precipitation is well represented by ERA5 over the Sierra Nevada, in agreement with previous studies highlighting the good performances of ERA5 precipitation especially in extratropical regions (Lavers et al., 2022). We find an overestimation of snow accumulation in high elevation which occurs only above 3000 m asl. In the study domain, the maximum elevation of ERA5 and ERA5-Land grid cells are 2654 m and 3100 m respectively. Hence the overestimation shown in Figure 5 is likely     due to the extrapolation of ERA5 precipitation by MicroMet. MicroMet uses monthly coefficients to adjust precipitation with elevation. These coefficients were derived from a large precipitation gauge dataset in the Western North America including the Tuolumne river catchment (Liston and Elder, 2006b). As a result, they only represent a first order variation of precipitation with elevation and may introduce large biases only in areas whose fine scale elevation (i.e. at the scale of the 100 m grid) deviates substantially from the ERA5 grid cell elevation. A possible source of error in high elevation regions is the lack of gravitational transport in SnowModel. High elevation and steep slopes are prone to avalanches thereby reducing the accumulated snow in these areas during the winter season (Quéno et al., 2023). However, we did not find a clear correlation between the terrain slope and the model error (Fig. 5). Slopes above 15% have a slightly wider error distribution but the mean absolute biases remain below 0.10 m w.e for both simulations. We also verified the residuals distribution by average slope classes computed from a 3 m resolution slope raster (computed from the ASO snow-off lidar DEM) and found similar results (see Figure A2 of the appendix). Hence, we do not see clear evidence that the lack of gravitational transport is the main cause of the high elevation biases. Another significant source of uncertainty is related to the albedo parameterization in SnowModel. The deposition of light absorbing particles like dust can reduce albedo and therefore increase melt especially at high elevation (Skiles et al., 2018; Dumont et al., 2020). This might explain the relative increase of the SWE bias between the 1st of April and the 27th of May at all elevations above 2500 m (Figure 5).

At catchment scale we do not find a clear difference between ERA5-SnowModel and ERA5-Land-SnowModel outputs. This suggests that the details of the downscaling scheme are not the primary factors of the simulation performance. However, there is a deviation between both simulations at high elevation. As shown in Figure 5, the downscaling of ERA5 creates a strictly increasing bias with elevation above 2500 m, whereas ERA5-Land creates a more complex bias that is negative between 2000 m and 3000 m and becomes positive above 3500 m. This more complex bias distribution reflects the fact that the output of the ERA5-Land SnowModel pipeline is the result of two downscaling schemes (first ERA5 to ERA5-Land, then ERA5-Land to 100 m using MicroMet, Fig. 2). ERA5-Land atmospheric variables are generated by linear interpolation of their ERA5 counterparts. ERA5-Land air temperature and humidity are also adjusted using the grid cell elevation using a daily lapse rate derived from ERA5 lower troposphere temperature vertical profile (Dutra et al., 2020). This is similar to the MicroMet algorithm. Yet, there are several differences. In particular, the air temperature downscaling scheme in ERA5-Land is based on a daily environmental lapse rate derived from ERA5 lower troposphere temperature vertical profiles (Muñoz Sabater, 2019), whereas MicroMet lapse rates are fixed by month. Unlike ERA5-Land, MicroMet also adjusts the precipitation rates using a function of elevation (Liston and Elder, 2006b). This is the cause of the non-monotonic evolution of the SWE bias by elevation from ERA5-Land-SnowModel. In future applications we will favor ERA5 instead of ERA5-Land to avoid conflicting processes in the downscaling of atmospheric variables. It makes it easier to adjust the precipitation correction factors from local data. Using ERA5 is also more practical as it significantly reduces the download time, computing cost and memory usage of our pipeline.

In Figure 3, we note the very good performance of ERA5-Land SWE at catchment scale despite its coarse scale (9 km resolution). This result is in line with Muñoz-Sabater et al. (2021) who find better performances of ERA5-Land than ERA5 between 1500 m and 3000 m a.s.l. because 68% of the Tuolumne River catchment is in this elevation band. Shao et al. (2022) found a similar accuracy of the ERA5-Land SWE dataset with an RMSE below 0.04 m w.e. in regions north of 45°N. This evaluation was performed using point-scale in situ measurements over large flat regions and not in complex mountain terrain like the Tuolumne Basin where the high spatial variability of SWE makes such evaluation more challenging (Mortimer et al. 2024). Overall, the performance of ERA5-Land SWE needs to be consolidated in other regions and ideally over larger domains of mountainous areas. Previous studies suggested that a resolution below 500 m is required to properly simulate the snowpack distribution (Baba et al., 2019; Bair et al., 2023). In addition, ERA5-Land resolution does not meet the essential climate variable requirements set by the World Meteorological Organization for SWE (goal is 500 m resolution) (WMO e-Library, 2024).

Regarding Sentinel-1, Figure 7 suggests that the snow depth is well captured by the C-SNOW algorithm at 1 km resolution. Although we are interested in SWE and not snow depth, the ASO program has shown that useful SWE products can be derived from remotely sensed snow depth when combined with in situ measurements and modeled snow density (Painter et al., 2016). Figure 7 shows that Sentinel-1 snow depth dataset agrees moderately with the spatial variability inside the catchment,

although we note a slight underestimation for all three dates before the melting period (2017 and 2019) and after it (2018). There is no clear pattern in the errors that emerge from these three dates. Other studies highlighted that the C-SNOW algorithm is not adapted to retrieve snow depth of shallower snowpack (<1.5 m) (Broxton et al., 2024; Hoppinen et al.,2024) which could be a significant obstacle for an operational use of this product. The modeling approach with ERA-5 (Land) and SnowModel yields similar performances in terms of snow depth as the C-SNOW product on the same dates. However, two patterns appear on Figure 7 for these approaches. i) The simulations with ERA5 and SnowModel are mostly centered around a negative bias constant with the observed snow depth before the melting period (2017 and 2019), probably representing a small negative bias in the ERA5 precipitation. ii) The simulations with ERA5-Land SnowModel seem to cap at 4 m which could be the result of the two consecutives downscaling in the precipitations : the combination of an underestimation of ERA5 precipitation and its downscaling, plus the limitation of the elevation difference between ERA5-Land stations and the DEM so the MicroMet precipitation factor cannot enhance enough the high resolution precipitations. Overall, the key difference in the Tuolumne catchment is that the model provides temporally continuous SWE, snow depth and other relevant variables like snowmelt runoff, whereas C-SNOW snow depth products are temporally sparse and often masked during the melt season.

Our study has several limitations. Despite the large amount of data that were used for this study, our analysis is biased towards the melt season since most of the ASO surveys were performed during the melt season for operational purposes. As a consequence, the evaluation of the Sentinel-1 snow depth is limited to three dates only. In addition, we used ASO SWE which is not a direct observation but a combination of accurate snow depth measurements and modeled snow density. Previous work has shown that SWE variability is mostly driven by the snow depth variability (López-Moreno et al., 2013; Sturm et al., 2010). Another limitation is the fact that ERA5 meteorological forcings may not be homogeneous across the globe due to the uneven distribution of the assimilated observations. In addition, MicroMet precipitation correction coefficients were obtained from a large region covering the study area, hence they may not be applicable in other regions. Therefore, we cannot generalize our results to other regions. However, the increasing weight of global satellite observations in ERA5 over time suggests that ERA5 performances should be more spatially homogeneous in the recent and upcoming years. As a consequence, ERA5 uncertainty varies with time since more and more data are available for data assimilation (Bell et al., 2021). This could be a limitation to compute trends over large periods (Bengtsson et al., 2004).

However these errors have a low impact at the catchment scale and we can conclude that ERA5-SnowModel is promising for water resources applications. This pipeline can be used to simulate SWE in near real time without the need of in situ measurements. The development of a parallel version of SnowModel opens the door to continental scale applications (Mower et al., 2023).

## 5 Conclusion

We have evaluated a pipeline to simulate the snowpack in mountainous catchment from global datasets only. This tool is based on Copernicus land cover and DEM, ERA5 (or ERA5-Land) and SnowModel. It uses SnowModel/MicroMet to downscale meteorological variables from ERA5 before computing accumulation and ablation processes using other SnowModel submodels. It can generate daily gridded snow water equivalent over any region and any period of interest since 1940. Based on 49 reference SWE surveys spanning seven contrasted hydrological years, we find that the ERA5-SnowModel combination simulates well the SWE at the scale of the Tuolumne river catchment, with RMSE of 0.06 m (and 0.08 m with ERA5-Land) and correlation of 0.99 (with both datasets). The SWE is also well simulated by elevation bands, except in the highest elevation band where unrealistic SWE values were simulated. Between ERA5 and ERA5-Land, ERA5 is more convenient to use especially because it requires less computing resources. Using the near-real-time release of ERA5 allows the simulation of SWE with a 5 day latency. This makes this method usable in operational context and competitive with a satellite-based approach. In particular, we found that it simulates the snow depth as well as the C-SNOW products derived from Sentinel-1, which is only available during dry snow conditions.

Our study focused on a single catchment due to the availability of the ASO SWE products. However, ERA5 skills may vary geographically and temporally due to the heterogeneity of assimilated data sources. Therefore, the performance of this method should be evaluated in other mountain catchments. Recent remote sensing methods to retrieve snow depth from very high resolution stereoscopic imagery will be useful for that perspective. To further reduce the errors in the simulation at finer resolution, we also intend to add a data assimilation module in order to take advantage of other global datasets such as the snow cover area from remote sensing.

**Competing Interest**

Co-authors KB was a member of the NASA ASO team (which produced the lidar data used in this study). KB is currently employed by ASO, Inc., formed as a result of the ASO NASA technology transition effort.

**Acknowledgements**

We sincerely thank G. Liston for sharing the SnowModel code. We thank Franziska Koch and Olivier Merlin for fruitful discussions about this work.

**Code Availability**

The wrapper around the SnowModel code can be found here : SOURP Laura / ERA_SnowModel_Pipeline · GitLab: https://src.koda.cnrs.fr/laura.sourp.1/era_snowmodel_pipeline, last access: 15 March 2024.

394

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

 **Appendix**

| Copernicus class number | Copernicus Vegetation type | Forest type | Leaf type | Chosen corresponding SM class | SM class number |
|---|---|---|---|---|---|
| 0 | Nodata | | | | -9999 |
| 20 | Shrubs | | | Mesic upland shrub | 6 |
| 30 | Herbaceous Vegetation | | | Grassland rangeland | 12 |
| 40 | cropland | | | short crops | 23 |
| 50 | Urban | | | Residential/urban | 21 |
| 60 | sparse vegetation | | | Bare | 18 |
| 70 | Snow and ice | | | Permanent snow/glacier | 20 |
| 80 | Permanent water bodies | | | water/ possibly frozen | 19 |
| 90 | Herbaceous wetland | | | Shrub wetland/ riparian | 9 |
| 100 | Moss and lichen | | | Bare | 18 |
| 111 | closed forest | evergreen | needle | Coniferous forest | 1 |
| 112 | closed forest | evergreen | broad | Coniferous forest | 1 |
| 113 | closed forest | deciduous | needle | Deciduous forest | 2 |

| 114 | closed forest | deciduous | broad | Deciduous forest | 2 |
|-----|---------------|-----------|-------|------------------|---|
| 115 | closed forest | mixed | | Mixed forest | 3 |
| 116 | closed forest | unknown | | Mixed forest | 3 |
| 121 | open forest | evergreen | needle | Coniferous forest | 1 |
| 122 | open forest | evergreen | broad | Coniferous forest | 1 |
| 123 | open forest | deciduous | needle | Deciduous forest | 2 |
| 124 | open forest | deciduous | broad | Deciduous forest | 2 |
| 125 | open forest | mixed | | Mixed forest | 3 |
| 126 | open forest | unknown | | Mixed forest | 3 |
| 200 | open sea | | | Ocean | 24 |


**Table A1 : Correspondence table between Copernicus land cover and SnowModel vegetation classes**

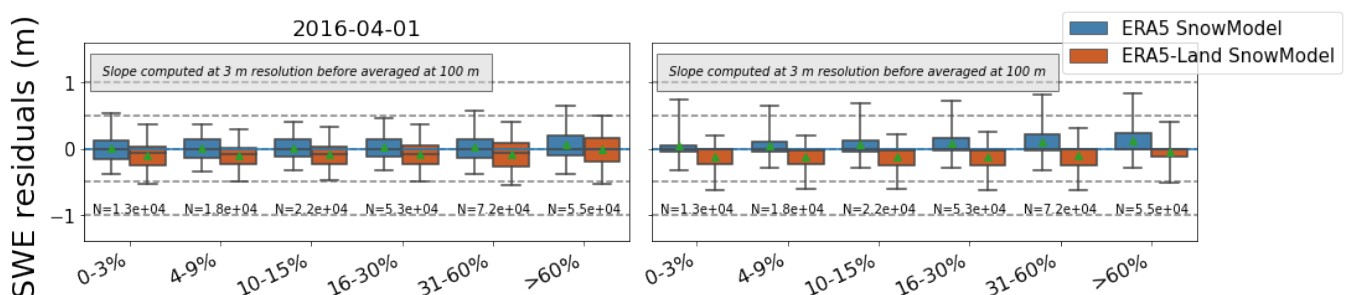


**Figure A2: Distribution of the residuals between the SnowModel simulated SWE and the ASO SWE at 100 m resolution**
**in the Tuolumne river catchment (in m w.e.) on the 1st of April 2016 (left) and the 27th of May (right), stratified by**

slope. Whiskers show the 5-95 percentile, the line in each box represents the median of the distribution and the green triangle shows the mean. Slope has been calculated using the DEM at 3 m resolution and has been resampled with an average algorithm at 100 m.