# Peer review of "Evaluation of high resolution snowpack simulations from global"

_EGUsphere, 2024_

## Author Comment (AC1)

The paper shows some nice comparisons between model results (ERA5-Land combined with SnowModel) and measurements from the Airborne Snow Observatory. Given the more discouraging conclusions about such models by Liu et al. (2022), I am surprised, but the analysis here seems robust. As the Conclusion notes, the analysis provides a viable method to estimate the water resources in the snowpack in areas with only an austere information infrastructure.

We would like to thank Jeff Dozier for taking the time to review our manuscript. We are confident that we can address every comment in a revised manuscript as explained below.

A few comments to improve the manuscript:

Line "06" (106?, bottom of section 2.1). The reported accuracy of SWE, <0.01 m, is ambitious. In their reports to the water agencies, ASO quotes a density RMS uncertainty of ±20 $kg/m^3$, but verification is based on snow courses and snow pillows, which are all on open, flat terrain and have their own uncertainties of SWE and depth.

Especially, you should note that ASO's translation of snow depth to SWE depends on local measurements of density, typically snow pillows that have a depth sensor also along with snow courses where both SWE and depth are measured.

The reported accuracy on the 3 m snow depth products is 0.08 m(Painter et al., 2016) and from spatially intensive sampling, the reported accuracy for the 50m snow depth products is < 0.01 m (Painter et al., 2016, Figure 15). There are no published references for the 50 m SWE product. However, for a 1m deep snowpack and a conservative 10% uncertainty in snow density (20-50 kg/m3), we estimate the uncertainty of the 50m SWE products to be 0.02 - 0.05 m w.e.

Section 2.2.2: How are you getting snow albedo for the EnBal part of SnowModel? The ASO spectrometer can be used to retrieve values, but the combined ERA-Land/SnowModel uses the ASO data for validation, not as a driver. The melt rate and disappearance date of the snowpack are sensitive to albedo and consequent radiative forcing by light-absorbing particles (Painter et al., 2010).

We used the default values implemented in EnBal (Liston and Elder, 2006). The default value of the snow cover albedo is 0.8 in dry conditions. The default value of melting snow albedo is 0.45 under the forest canopy and 0.60 in non-forested areas.

Figure 3: The colors used to identify the lines in the plots are too indistinct. Perhaps combine color with line style to make the differences more obvious?

We propose to update figure 3 with this design

[Figure]

Figure 6: Label the axes. They appear to be UTM zone 11N coordinates, but the identification of the comparison in rotated text is confusing. At first I thought they had something to do with the y-axis.

We have added the axes labels as suggested.

[Figure]

Figure 7 Line 47 in caption: recommend data "are" instead of "is".

ok

Line 01 in the Discussion. The phrase "the ASO program has shown that useful SWE products can be derived from remotely sensed snow depth" needs a caveat, in that the ASO model of snow density is adjusted based on in situ measurements of snow density.

*We agree with this relevant comment. We will change the sentence to "Although we are interested in SWE and not snow depth, the ASO program has shown that useful SWE products can be derived from remotely sensed snow depth when combined with in situ measurements and modeling of snow density"*

Line 21-22 in the Discussion. Perhaps cite the Liu et al. (2022) analysis here?

*We agree the Liu et al. (2022) analysis should be cited. We added the citation in the introduction :  However, reanalyses cannot be used directly to force a mountain snowpack model because the grid cell size is too coarse (approximately 30 - 50 kilometers for ERA5 and MERRA-2 respectively), which creates large biases in the computed SWE (Wrzesien et al., 2019; Liu et al.,2022).*

*Line 21-22 in the Discussion refers to the meteorological forcings of ERA5 used in the study (while Liu et al. (2022) focused on direct SWE products from the reanalyses). We will rephrase with : Another limitation is the fact that ERA5 meteorological forcings may not be homogeneous across the globe due to the uneven distribution of the assimilated observations.*

I agree with the final paragraph of the Discussion. The combination of ERA5, SnowModel, and Sentinel-1 provides a way to analyze the snowpack in mountains with only an austere infrastructure. There are uncertainties of course, but the methods could provide some information in areas where few data exist.

Support for Open Science: The manuscript should identify the sources of data and code availability used in the analyses. I could do my own searches, but statements like "from the Copernicus Climate Change Service (C3S) and can be queried via their application programming interface" (Line 92) could be phrased more helpfully.

*We will add a reference to the tutorials of the Climate Change Service on how to retrieve the data: (Retrieving data — Climate Data Store Toolbox 1.1.5 documentation, 2024)*

Similarly, the citation to "Copernicus Digital Elevation Model, 2023" (Line 96) is not in the bibliography.

*The reference is actually the second item of the bibliography.*

Some information is missing about the "code availability section" mentioned on Line 45.

*We added a code availability section at the end of the paper.*

**References**

Retrieving data — Climate Data Store Toolbox 1.1.5 documentation: https://cds.climate.copernicus.eu/toolbox/doc/how-to/1_how_to_retrieve_data/1_how_to_retrieve_data.html, last access: 27 June 2024.

Liston, G. E. and Elder, K.: A distributed snow-evolution modeling system (SnowModel), J. Hydrometeorol. 76 1259-1276, 2006.

Liu, Y., Fang, Y., Li, D., and Margulis, S. A.: How Well do Global Snow Products Characterize Snow Storage in High Mountain Asia?, Geophys. Res. Lett., 49, e2022GL100082, https://doi.org/10.1029/2022GL100082, 2022.

Painter, T. H., Berisford, D. F., Boardman, J. W., Bormann, K. J., Deems, J. S., Gehrke, F., Hedrick, A., Joyce, M., Laidlaw, R., Marks, D., Mattmann, C., McGurk, B., Ramirez, P., Richardson, M., Skiles, S. M., Seidel, F. C., and Winstral, A.: The Airborne Snow Observatory: Fusion of scanning lidar, imaging spectrometer, and physically-based modeling for mapping snow water equivalent and snow albedo, Remote Sens. Environ., 184, 139–152, https://doi.org/10.1016/j.rse.2016.06.018, 2016.

Painter, T. H., Deems, J. S., Belnap, J., Hamlet, A. F., Landry, C. C., and Udall, B.: Response of Colorado River runoff to dust radiative forcing in snow, Proceedings of the National Academy of Sciences, 107, 17125-17130, https://doi.org/10.1073/pnas.0913139107, 2010.

---

## Author Comment (AC2)

**General comments**

The authors present a comprehensive evaluation of high-resolution snowpack simulations forced with globally available datasets, in particular coarse resolution meteorological data downscaled to the model grid. Thus, the study showcases a generic tool for performing snow cover simulations in any region of the world efficiently and with low effort. The simulations presented in the study, performed for the Tuolumne River catchment (Sierra Nevada, USA), were evaluated against high-resolution snow water equivalent (SWE) data derived from Lidar measurements of snow depth and modelled bulk snow densities. The simulations show promising results with comparable performance as satellite-derived snow characteristics for the study basin. In contrast to the remote sensing observations, the snow model results are always available, which is a significant advantage over the occasional satellite retrievals.

Overall, appropriate methods are used in the study and the results are relevant and promising. However, the presentation and discussion of the results sometimes lacks clarity and depth in my opinion. The description of the results deserves a few more details, whereas the discussion requires stronger links to the results themselves (foremost by including more references to specific figures). Furthermore, the paper should likely also be improved language-wise, preferably by a native English speaker. In spite of the shortcoming listed above, the paper is pleasant to read, contains a wealth of interesting results and is a valuable contribution to the snow modelling community. Detailed comments are listed below.

We thank the reviewer for the careful evaluation of our work. We appreciate the positive comments and relevant suggestions. We can implement every suggestion in a revised manuscript as detailed below.

**Specific comments**

Page 1, line 13: Consider changing "sourcing" to using and "climate" to "meteorology".

ok

Page 1, line 18: Change from "snow depth to Sentinel-1 snow depth retrievals" to "snow depth to Sentinel-1 retrievals".

ok

Page 1, abstract: The concluding sentence of the abstract should be improved. One option would be to add a sentence stating directly that the snow model provides results anywhere at anytime in contrast to satellite retrievals.

We propose to reformulate the last sentence:

However, Sentinel-1 snow depth products are sparse and often masked during the melt season, whereas ERA5-SnowModel provides spatially and temporally continuous SWE.

Page 2, line 34: Please also cite Lievens et al. (2022) and adapt the sentence accordingly.

ok

Page 2, line 46: Include the missing "have": "There reanalyses have also…"

ok

Page 2, lines 59-60: The sentence "However, the evaluation of these simulations relied on sparse in situ observations or MODIS snow cover area" seems incomplete. What is the drawback with these observations and why are more studies needed? Is it the coarse resolution of MODIS snow covered area?

Our intention was to highlight that these data do not allow to validate the spatial distribution of the snow depth or SWE across the landscape.

We propose to remove "However," to make the paragraph clearer:

*The evaluation of these simulations relied on sparse in situ observations or MODIS snow cover area (...) However, these in situ or remote sensing datasets did not allow a thorough evaluation of the model ability to capture snow mass across the landscape*

Page 3, lines 68-79: Consider adding the spatial resolution of the model simulations already here.

ok

Page 5, lines 00-01: Please mention the physical reason why the satellite retrievals do not provide data during the snowmelt period and add a reference supporting the statement.

We propose to add the following sentence and references:
*When the snowpack is wet, there is a larger absorption and reflection of the microwave signal emitted by Sentinel-1 which greatly decreases the performances of the C-SNOW algorithm (Lievens et al., 2019; Tsai et al., 2019).*

Page 5, line 06: Important, the statement "…50 m SWE is less than 0.01 m w.e" needs a reference.

The reported accuracy on the 3 m snow depth products is 0.08 m (Painter et al., 2016) and from spatially intensive sampling, the reported accuracy for the 50m snow depth products is < 0.01 m (Painter et al., 2016, Figure 15). There are no published references for the 50 m SWE product. However, for a 1m deep snowpack and a conservative 10% uncertainty in snow density (20-50 kg/m3), we estimate the uncertainty of the 50m SWE products to be 0.02 - 0.05 m w.e

Page 5, line 15: What is "grassland rangeland"?

It is the SnowModel class name for herbaceous vegetation (graminoids and forbs).

Page 7, line 40: Consider changing from "Appendix Table A1" to "see Table A1 in appendix".

ok

Page 7, line 58: Consider changing to "very coarse resolution of approximately 31 and 9 km (Fig. 1 and 2)".

ok

Page 7, lines 62-63: Consider changing to "…the snow depths given by ASO, Sentinel-1, and ERA-SnowModel were…".

ok

Page 8, lines 65-66: Please reformulate these two sentences. The second sentence needs to reference the first, otherwise it is not clear for what the performance metrics were computed.

We propose the following reformulation : *We computed the distributed residuals by subtracting the ASO snow depth from both SnowModel simulations and Sentinel-1 data. We averaged the residuals to compute the bias for each date. We also computed the standard deviation of the error and the RMSE over the catchment for each date .*

Page 8, line 76-78: Please reformulate the sentence. It is too long and hard to read.

ok

Figure 3: Consider using dashed lines for ERA5 and ERA5-Land.

ok

Page 9, lines 84-85: It is likely not needed to describe the lines here since this information is already provided in the legend of the figure.

ok

Page 9, line 89: The sentence "Considering the entire simulation period, 10% of the cells have an RMSE above 0.5 m w.e." seems somewhat misplaced and is hard to understand.

This is the transition between the catchment scale analysis to the pixel scale analysis.. It will be rephrased in the paper with : *We computed a map of RMSE using all the 49 validation dates we have between 2013 and 2019. 10% of the cells in this map have a RMSE above 0.5 m w.e*

Page 10, lines 1-2: Why were these two dates selected for the analysis?

We propose to clarify this point in the revised manuscript as follows:

*We aimed to distinguish the model performance in terms of accumulation and ablation processes to better separate the sources of uncertainties in future studies. Therefore we selected a date before the melting season (April 01) and a date near the end of the melting season (May 27).*

Figure 5, caption: Why is the second date not mentioned in the caption?

This was an oversight. We will correct this.

Page 11, line 14: Is "mean residuals" the same as bias?

yes it is. It will be rephrased with "*mean of residuals (bias)*" in the revised manuscript.

Page 11, line 25: Consider changing to "…resolution using upscaled ASO…".

ok

Page 11, lines 28-29: What does "these missing values are propagated at 1 km resolution" mean?

We resample the ASO products by averaging all pixels inside a square cell of 1 km. If there is at least one missing value among the contributing pixels, a missing value is attributed to the target 1 km cell.

Page 11, line 30: Is not the exact area used between the methods or the dates, or both?

Both : all the pixels shown in figure 6 are taken into account in Table 1. For ERA5-SnowModel, the mask is the same for the three dates because the missing values are due to the missing values in the ASO data. With Sentinel-1, the missing values are due to i) the missing values in the ASO and ii) the missing values in the Sentinel-1 algorithm.The second one are time dependent and therefore the statistics in Table 1 are not computed on the same area from one date to another, nor on the same area as ERA5-SnowModel.

Figure 7: Consider merging Table 7 into this figure by including texts with the statistics. For an example of what I propose, see Figure 5 in Fontrodona-Bach et al. (2023). The scatter plots could potentially also be improved by showing the scatter density, just like the left panels in the Figure 5 by Fontrodona-Bach et al. (2023).

We will merge Table 1 and Figure 7 as suggested. However, the numbers of points are not sufficient to make nice density plots (2D histograms). It would add unnecessary information (colorbar) and decrease the readability of the plots.

Page 13, line 53: What discontinuities in ERA5 SWE? Are these visible in Figure 3?

They are not visible in Figure 3. There are some discontinuities in the ERA5 SWE appearing in 1976 due to the implementation of new snow depth products into the ERA5 assimilation scheme. When these products are assimilated, ERA5 caps the snow depth data at 1.4 m to avoid an overestimation of the snow depth (personal communication from Patricia de Rosnay, ECMWF). This creates a strong discontinuity in the ERA5 snow time series (see figure below). Because the meteorological forcings would not be impacted by this threshold on snow, using this pipeline could be a way to bypass this discontinuity. However, other meteorological variables in ERA5 might also be affected by the growing number of data assimilated (Bengtsson et al., 2004).

[Figure]

Page 14, line 58-59: Please improve the language of the sentence "We find an overestimation of snow accumulation in high elevation however which occurs only above 3000 m asl".

We suggest to reformulate:

*We find an overestimation of snow accumulation at high elevations, specifically occurring above 3000 m asl.*

Page 14, lines 66-67: Avalanches move snow from higher to lower altitudes but does not reduce snow amounts. Please rephrase the sentence.

*High elevation and steep slopes are prone to avalanches thereby reducing the accumulated snow in these areas during the winter season (Quéno et al., 2023)*

Page 14, lines 75-77: Please refer to Figure 5.

ok

Overall, as mentioned in the general comments, provide more links in the discussion to results by adding appropriate cross-references to figures and tables.

We will follow this suggestion in the revised manuscript with : *This result is in line with Muñoz-Sabater et al. (2021) who find better performances of ERA5-Land than ERA5 between 1500 m and 3000 m a.s.l. because 68% of the Tuolumne River catchment is in this elevation band.*

Page 15, lines 91-93: The sentence is formulated awkwardly. What does "carries 68 % of the Tuolumne River catchment" mean?

It meant that 68% of the catchment has an elevation between 1500 m and 3000 m. Rephrased in the new manuscript

Page 15, lines 1-2: This statement requires at least one reference.

*We will add this reference to the sentence in the manuscript : (Margulis et al., 2019)*

Page 15, line 6: What is hard to understand about the error patterns of Sentinel-1 compared to the other methods?

*Figure 7 shows that Sentinel-1 snow depth dataset seems to represent quite accurately the spatial variability inside the catchment, although we note a slight underestimation for all three dates before the melting period (2017 and 2019) and after it (2018). There is no clear pattern in the errors that emerge from these three dates.The modeling approach with ERA-5 (Land) and SnowModel yields similar performances in terms of snow depth as the C-SNOW product on the same dates. However, two patterns appear on Figure 7 for these approaches. i) The simulations with ERA5 and SnowModel are mostly centered around a negative bias constant with the observed snow depth before the melting period (2017 and 2019), probably representing a small negative bias in the ERA5 precipitation. ii) The simulations with ERA5-Land SnowModel seem to cap at 4 m which could be the result of the two consecutives downscaling in the precipitations : the combination of an underestimation of ERA5 precipitation and its downscaling, plus the limitation of the elevation difference between ERA5-Land stations and the DEM so the MicroMet precipitation factor can not enhance enough the high resolution precipitations*

Page 15, lines 11-14: What has the first part of the sentence about errors has to do with the second part about model differences? Please split this sentence into two, and improve the language.

*There are different error sources in the three methods which are neither insignificant nor prohibitive for an operational use. The key difference is that the model provides temporally continuous SWE, snow depth and other relevant variables like snowmelt runoff, whereas C-SNOW snow depth products are temporally sparse and often masked during the melt season.*

Page 16, lines 34-35: Consider providing a short description for each components of this tool since many readers start by reading the conclusions of a paper.

*It uses SnowModel/MicroMet to downscale meteorological variables from ERA5 before computing accumulation and ablation processes using other SnowModel submodels.*

Page 16, line 38: What does the "0.08 m" refer to?

*Indeed, this was not clear, we will reformulate as follows:*

*Based on 49 reference SWE surveys spanning seven contrasted hydrological years, we find that the ERA5-SnowModel combination simulates well the SWE at the scale of the Tuolumne river catchment, with RMSE of 0.06 m (and 0.08 m with ERA5-Land) and correlation of 0.99 (with both datasets)*

Page 16, lines 34-43: Example of paragraph that likely needs language improvements.

**Technical comments**

Page 3, line 70: Misplaced white space in 50 m.

ok

Page 7, line 39: Missing whitespace.

ok

Page 7, line 56: Missing comma after additionally.

ok

Page 15, line 92 and 93: Wrong reference format.

ok

**References**

Fontrodona-Bach, A., Schaefli, B., Woods, R., Teuling, A. J., & Larsen, J. R. (2023). NH-SWE: Northern Hemisphere Snow Water Equivalent dataset based on in situ snow depth time series. Earth Syst. Sci. Data, 15(6), 2577-2599. https://doi.org/10.5194/essd-15-2577-2023

Bengtsson, L., Hagemann, S., and Hodges, K. I.: Can climate trends be calculated from reanalysis data?, J. Geophys. Res. Atmospheres, 109, https://doi.org/10.1029/2004JD004536, 2004.

Lievens, H., Demuzere, M., Marshall, H.-P., Reichle, R. H., Brucker, L., Brangers, I., de Rosnay, P., Dumont, M., Girotto, M., Immerzeel, W. W., Jonas, T., Kim, E. J., Koch, I., Marty, C., Saloranta, T., Schöber, J., and De Lannoy, G. J. M.: Snow depth variability in the Northern Hemisphere mountains observed from space, Nat. Commun., 10, 4629, https://doi.org/10.1038/s41467-019-12566-y, 2019.

Lievens, H., Brangers, I., Marshall, H. P., Jonas, T., Olefs, M., & De Lannoy, G. (2022). Sentinel-1 snow depth retrieval at sub-kilometer resolution over the European Alps. Cryosphere, 16(1), 159-177. https://doi.org/10.5194/tc-16-159-2022

Margulis, S. A., Fang, Y., Li, D., Lettenmaier, D. P., and Andreadis, K.: The Utility of

Infrequent Snow Depth Images for Deriving Continuous Space-Time Estimates of Seasonal Snow Water Equivalent, Geophys. Res. Lett., 46, 5331–5340, https://doi.org/10.1029/2019GL082507, 2019.

Tsai, Y.-L. S., Dietz, A., Oppelt, N., and Kuenzer, C.: Remote Sensing of Snow Cover Using Spaceborne SAR: A Review, Remote Sens., 11, 1456, https://doi.org/10.3390/rs11121456, 2019.

---

## Referee Report (RR1)

**November 14, 2024**

Evaluation of high resolution snowpack simulations from global datasets and comparison with Sentinel-1 snow depth retrievals in the Sierra Nevada, USA
By: L. Sourp et al.

**General Comments**

In this paper, the authors create a modeling pipeline which leverages global-scale meteorological analyses (ERA5 and ERA5-Land) to force SnowModel and produce 100 m SWE estimates in the Tuolumne River Basin for a seven year study period. They then compare the modeled results to ASO lidar SWE/snow depth and S1-derived snow depths, showing surprisingly performance even with the coarse scale forcing data. The results presented here are novel/robust and provide a baseline to extend this modeling technique to other data sparse regions around the globe, which could provide marked enhancements to SWE/water resource forecasting.

The manuscript is clearly written and the comments from the previous reviewers have all been properly addressed and integrated into the text. I find the discussion of the results well throughout and the figures well presented. The comments provided below are mostly minor, the most important being the adding of numerical error statics to the paragraph between L233–245. Once these are addressed this article is suitable for publication and will be a solid edition to TC. Congrats on a very nice study!

-Jack Tarricone

**Specific Comments**

L32: How does Pléiades retrieve snow depth? I know, but a bit more information on stereo photogrammetry would be good for a broader audience. Same thing for ICESat-2.

L33: Add a bit of info on co/cross pol S1 retrieval algorithm

L70: "worldwide" is a bit confusing, as it reads on first pass this SWE data is worldwide. Maybe "Globally publicly available from this basin" ?

L113: I like the discussion of the density uncertainty here, but would add a reference to Raleigh & Small (2017) and rework for a more robust statement.

L149–153: I recommend creating a table in the appendix with all model parameters mentioned in this text so future work attempting to replicate your work knows exactly what you did. While I see they're buried in the Github page, I'm not sure exactly where to find them.

L196–202: A bit confused on how you're referencing "interquartiles" here. Also the text states, "...for the wet year 2017, they peak respectively at 0.64 and 0.82 m." Yet, when I look at the boxplots it seems 06-04 has the max value, which barely extends below –0.5.

Please check this paragraph for clarity and correct numbers.

L214: Note TC date formatting requirements (https://www.the-cryosphere.net/submission.html): "1 April 2016", I won't ask you to do this but will likely need to be updated in the copy editing stage!

L225: Why is the resampling procedure set up this way? It seems like you're losing valuable information if you're throwing away a whole 1 km pixel if 1 of 400 50 m pixels is missing. Not saying it's incorrect but some justification of why this is the proper method should then be added in Section 2.2.3 then.

L233–245: Add values to bias, SD, R^2, and RMSE when referenced, this will likely require some tweaking of the language as well. You've performed solid analysis that is not being communicated clearly in this paragraph!

L237: Remove "seems to be" – no need for subjective language when you've conducted numerical analysis. Use the error metrics you calculated and state the performance of each dataset!

L240: How do we know S1 it underestimates? State specific metrics used to support this sentence.

Figure 7: Provide number of values in each scatter plot (n = xx). I only say this because you said there are different numbers in each, so the reader should know how much that varies.

L273: I would add a figure in the appendix of this analysis, as you're referencing something you did but provide no data/figure to back it up.

L300: I would add some context to the Shao et al. RMSE of 0.04 m for ERA5-Land. What are some of the uncertainties associated with validating a 9 km pixel against point-based in situ observations? Would these be magnified in complex mountain terrain?

L309: Not sure I totally agree here, "seems to represent quite well" yet R^2 0.25–0.53. Maybe "agrees moderately"?

L319: Recent work has shown S1 struggles in shallow snow (<1.5 m), as there is almost no physical co/cross pol backscattering signal detectable (Broxton et al., 2024; Hoppinen et al., 2024). The technique has been shown to work well in the Alps and moderately well here as they both have deeper snowpacks. I would caution against recommending it for operational use as (1) Many snowpacks are not deep and therefore not well suited, (2) No one has been able to produce the Lievens method to anywhere near the quality of the closed-source code he has. This supports that your modeling pipeline is superior!

**Technical Comments:**

Fig 1: Change color scale to 0.

L97: Replace '(see below)' with the specific section you're referencing.

L100: Add link to C-SNOW website here.

L113: What does "w.e." mean here and throughout the manuscript? Excuse my ignorance if this is a common phrase.

L132: DEM already defined.

L199: Noting "w.e." again. Found a few examples of what it could but still unsure.

L210: State two dates.

Figure 5: Added (left) and (right) in caption after corresponding dates.

L212: Remove double period.

L263: "most probably" -> likely

L308: "modelisation of snow density" -> modeled snow density

L341: "global datasets only" ->  global publicly available atmospheric reanalysis datasets only

L349: "near real time" -> near-real-time

**Bibliography:**

Broxton, P., Ehsani, M. R., & Behrangi, A. (2024). Improving Mountain Snowpack Estimation Using Machine Learning With Sentinel-1, the Airborne Snow Observatory, and University of Arizona Snowpack Data. *Earth and Space Science*, *11*(3), e2023EA002964. https://doi.org/10.1029/2023EA002964

Hoppinen, Z., Palomaki, R. T., Brencher, G., Dunmire, D., Gagliano, E., Marziliano, A., et al. (2024). Evaluating Snow Depth Retrievals from Sentinel-1 Volume Scattering over NASA SnowEx Sites. *EGUsphere*, 1–35. https://doi.org/10.5194/egusphere-2024-1018

**(**^ published article should be coming out soon so be on the lookout for that)

Raleigh, M. S., & Small, E. E. (2017). Snowpack density modeling is the primary source of uncertainty when mapping basin-wide SWE with lidar: Uncertainties in SWE Mapping With Lidar. *Geophysical Research Letters*, *44*(8), 3700–3709. https://doi.org/10.1002/2016GL071999

---

## Author Response (AR2)

**November 14, 2024**
Evaluation of high resolution snowpack simulations from global datasets and comparison with Sentinel-1 snow depth retrievals in the Sierra Nevada, USA
By: L. Sourp et al.

Dear Editor

We replied below to the third reviewer who provided many relevant suggestions. We also noted that after the second round of review, you suggested to "clearly and more explicitly link the discussion to specific results (...) Overall, try to avoid the reader asking him/herself: "So what?". We believe that the modifications made in response to the reviewers should answer this recommendation as well. In addition, we highlight that we wrote a clear statement in the end of the discussion "*we can conclude that ERA5-SnowModel is promising for water resources applications. This pipeline can be used to simulate SWE in near real time without the need of in situ measurements.*"

Many thanks again for handling our manuscript,

Laura Sourp and Simon Gascoin on behalf the co-authors.

**General Comments**

In this paper, the authors create a modeling pipeline which leverages global-scale meteorological analyses (ERA5 and ERA5-Land) to force SnowModel and produce 100 m SWE estimates in the Tuolumne River Basin for a seven year study period. They then compare the modeled results to ASO lidar SWE/snow depth and S1-derived snow depths, showing surprisingly performance even with the coarse scale forcing data. The results presented here are novel/robust and provide a baseline to extend this modeling technique to other data sparse regions around the globe, which could provide marked enhancements to SWE/water resource forecasting.

The manuscript is clearly written and the comments from the previous reviewers have all been properly addressed and integrated into the text. I find the discussion of the results well throughout and the figures well presented. The comments provided below are mostly minor, the most important being the adding of numerical error statics to the paragraph between L233–245. Once these are addressed this article is suitable for publication and will be a solid edition to TC. Congrats on a very nice study!

-Jack Tarricone

We would like to sincerely thank Jack Tarricone for the positive comments on our work and the very useful review. We have addressed every specific comment in a revised manuscript as explained below.

**Specific Comments**

L32: How does Pléiades retrieve snow depth? I know, but a bit more information on stereo photogrammetry would be good for a broader audience. Same thing for ICESat-2.

We have added a sentence for both methods to explain a bit more how they work (L33-35). *"Pléiades **very high resolution stereoscopic images can be used to generate snow depth images by differencing two digital elevation models** (...) ICEsat-2 **lidar altimetry** has the potential to provide snow depth data at global scale but with a sparse sampling".*

L33: Add a bit of info on co/cross pol S1 retrieval algorithm

We expanded the description of the Sentinel-1 algorithm: *"This method, **which is based on an empirical change detection method applied to the cross-polarization ratio** (...)"*

L70: "worldwide" is a bit confusing, as it reads on first pass this SWE data is worldwide. Maybe "Globally publicly available from this basin" ?

We agree that it could be confusing. We reformulated as follows : *"The ASO dataset on the Tuolumne catchment is the densest time series of high resolution snow depth (3 m) and SWE (50 m) maps **publicly available at this scale (1100 km²) in the world**."*

L113: I like the discussion of the density uncertainty here, but would add a reference to Raleigh & Small (2017) and rework for a more robust statement.

Many thanks for this relevant suggestion which strenghtens our uncertainty estimate. Based on the work of Rayleigh & Small (2017) we have revised our uncertainty estimate from 0.02-0.05 m w.e to 50 kg/m³. Indeed, Rayleigh & Small (2017) estimated an uncertainty in modeled density of 48 kg/m³ in the Tuolumne basin. Therefore, for a 1 m deep snowpack and an uncertainty in snow density of 50 kg/m³, the uncertainty of the 50 m SWE products is 0.05 m w.e. We note however that this uncertainty can be regarded as a conservative estimate as in situ measurements of snow density are also used by the ASO to adjust their density model (Painter et al., 2016). We modified the text accordingly L117-L121.

L149–153: I recommend creating a table in the appendix with all model parameters mentioned in this text so future work attempting to replicate your work knows exactly what you did. While I see they're buried in the Github page, I'm not sure exactly where to find them.

We agree that reproducibility is important. Apart from the parameters that we modified for this study and that we explicitly mentioned in the manuscript, there are many default parameters which would make a very large table. Therefore, we included the SnowModel configuration file which includes all the parameters and their description in our Github repository and modified the text as follows:

*We set all SnowModel parameters (the curvature length scale, curvature and wind slope weights, minimum wind speed, precipitations schemes for downscaling or for rain-snow fractions, subcanopy radiations schemes, various thresholds for wind transport calculations) to the default values (see the parameter file snowmodel.par in the code availability section)*

L196–202: A bit confused on how you're referencing "interquartiles" here. Also the text states, "...for the wet year 2017, they peak respectively at 0.64 and 0.82 m." Yet, when I look at the boxplots it seems 06-04 has the max value, which barely extends below –0.5. Please check this paragraph for clarity and correct numbers.

We are referring to the interquartile as the difference between the 75th percentile (upper quartile) and the 25th percentile (lower quartile). This means that we are not looking at the maximum value of a lower or upper quartile but the distance between these two quartiles. It is used as an indicator of the dispersion of the datasets. The numbers are correct : maximum interquartile with ERA5 in the 2017-05-02 with 0.64 m and with ERA5-Land it is the 2017-04-01 with 0.82 m. To clarify, we added this sentence: "*The spread of the residuals are shown with the interquartile (i.e., the difference between the 25 and 75th percentiles) inside the colored boxes, and with the 5-95th percentiles inside the whiskers.*"

L214: Note TC date formatting requirements https://www.the-cryosphere.net/submission.html): "1 April 2016", I won't ask you to do this but will likely need to be updated in the copy editing stage!

Thanks, we corrected it.

L225: Why is the resampling procedure set up this way? It seems like you're losing valuable information if you're throwing away a whole 1 km pixel if 1 of 400 50 m pixels is missing. Not saying it's incorrect but some justification of why this is the proper method should then be added in Section 2.2.3 then.

We chose this resampling method because we do not know if the missing data inside a 1km cell are evenly distributed among the values of the valid ASO HS 3 m pixels. We preferred losing some data that induce an incorrect bias. Also, most of the missing data are located at low elevation where there is almost no snow so this should not affect our statistics too much.

We propose this reformulation in section 2.2.4 : *We applied another validity mask for the cells where the snow depth is not always available to all three snow depth datasets (here representing 8.5% of missing data in the catchment). The missing values in the 3 m resolution ASO dataset are propagated at the 1 km resolution validity mask. This decreases the number of observations but ensures that the resampled 1 km snow depths maps are not biased by the spatial distribution of non-valid pixels in the 3 m ASO snow depth dataset.*

L233–245: Add values to bias, SD, R^2, and RMSE when referenced, this will likely require some tweaking of the language as well. You've performed solid analysis that is not being communicated clearly in this paragraph!
We propose the following reformulation :
*Figure 7 shows the Sentinel-1 observed and SnowModel simulated snow depth compared to the ASO observed snow depth, resampled to a 1 km resolution. On the 2017-03-03, Sentinel-1 has the lower bias (-0.43 m), standard deviation (0.86 m) and RMSE (0.96 m). These statistics are close to the ERA5-SnowModel simulations (respectively -0.49 m, 0.9 m, 1.02 m) while ERA5-Land-SnowModel simulations have a greater bias (-0.83 m) and RMSE*

*(1.2 m) with a comparable standard deviation (0.86 m). On the second date, the 2018-05-01, Sentinel-1 still performs the best with a bias of -0.05 m, and standard deviation and RMSE both equals to 0.21 m,. On this date, ERA5-Land-SnowModel simulations are similar to Sentinel-1 with a bias of -0.09 m, standard deviation of 0.26 m and RMSE of 0.27 m; while ERA5-SnowModel simulations underperform with a 0.16 m bias, a 0.41 m standard deviation and a 0.44 m RMSE.. Finally on the 2019-03-24, the closer data to the ASO snow depths is the ERA5-SnowModel simulations with an bias of -0.65 m, a standard deviation of 0.81 m and an RMSE of 1.04 m. Sentinel-1 data have the highest bias (-1.24 m) and RMSE (1.38 m), but the lowest standard deviation (0.61 m). ERA5-Land-SnowModel simulations also have a high bias (-0.92 m) and RMSE (1.17 m), with a standard deviation of 0.73 m.*

L237: Remove "seems to be" – no need for subjective language when you've conducted numerical analysis. Use the error metrics you calculated and state the performance of each Dataset!
We corrected it with "is", cf previous answer.

L240: How do we know S1 it underestimates? State specific metrics used to support this Sentence.
We change the sentence to : *In 2018, both the ASO and Sentinel-1 observed really low snow depths (<1 m) but there is still a negative bias (-0.05 m) in the Sentinel snow depth distribution*

Figure 7: Provide number of values in each scatter plot (n = xx). I only say this because you said there are different numbers in each, so the reader should know how much that varies.

That is right, here is the new version that we included in the revised manuscript.

[Figure]

**Figure 7: Scatter plots representing the observed and SnowModel simulated snow depth data as a function of ASO snow depth data, with a one to one line in black. All data are resampled at 1 km resolution. N is the number of values in each plot.**

L273: I would add a figure in the appendix of this analysis, as you're referencing something you did but provide no data/figure to back it up.
We agree, here is the figure that we added to the appendix :

[Figure]

**Figure A2: Distribution of the residuals between the SnowModel simulated SWE and the ASO SWE at 100 m resolution in the Tuolumne river catchment (in m w.e.) on the 1st of April 2016 (left) and the 27th of May (right), stratified by slope. Whiskers show the 5-95 percentile, the line in each box represents the median of the distribution and the green triangle shows the mean. Slope has been calculated using the DEM at 3 m resolution and has been resampled with an average algorithm at 100 m.**

L300: I would add some context to the Shao et al. RMSE of 0.04 m for ERA5-Land. What are some of the uncertainties associated with validating a 9 km pixel against point-based in situ observations? Would these be magnified in complex mountain terrain?

*This is a relevant question that we cannot answer easily. However, we can refer to a recent work by Mortimer et al. (2024) who have discussed this issue in the case of several reanalysis products including ERA5-Land.*

*Shao et al. (2022) found a **similar** accuracy of the ERA5-Land SWE dataset with an RMSE below 0.04 m w.e. in regions north of 45°N. This evaluation was performed using point-scale in situ measurements over large flat regions but not in complex mountain terrain like the Tuolumne Basin where the high spatial variability of SWE makes such evaluation more challenging **(Mortimer et al. 2024).***

L309: Not sure I totally agree here, "seems to represent quite well" yet R^2 0.25–0.53. Maybe "agrees moderately"?
*Yes agreed and changed accordingly.*

L319: Recent work has shown S1 struggles in shallow snow (<1.5 m), as there is almost no physical co/cross pol backscattering signal detectable (Broxton et al., 2024; Hoppinen et al., 2024). The technique has been shown to work well in the Alps and moderately well here as they both have deeper snowpacks. I would caution against recommending it for operational use as (1) Many snowpacks are not deep and therefore not well suited, (2) No one has been able to produce the Lievens method to anywhere near the quality of the closed-source code he has. This supports that your modeling pipeline is superior!
*Indeed these studies are relevant and could advise against an operational use in this context. We removed the sentence "There are different error sources in the three methods which are neither insignificant nor prohibitive for an operational use" and  added the following sentence above in the same paragraph "Other studies highlighted that the C-SNOW algorithm is not adapted to retrieve snow depth of shallower snowpack (<1.5 m) (Broxton et al., 2024; Hoppinen et al.,2024) which could be a significant obstacle for an operational use of this product. "*

**Technical Comments:**

Fig 1: Change color scale to 0.
*Changed to :*

[Figure]

L97: Replace '(see below)' with the specific section you're referencing.
We referenced the section 2.2.1 (Method/SnowModel) instead.

L100: Add link to C-SNOW website here.

L113: What does "w.e." mean here and throughout the manuscript? Excuse my ignorance if this is a common phrase.
It means meters of water equivalent. It has been added in the manuscript for clarification.

L132: DEM already defined.
We use DEM directly instead.

L199: Noting "w.e." again. Found a few examples of what it could but still unsure.

L210: State two dates.
The new sentence is Figure 5 shows the distribution of the residuals for two dates (2016-04-01 and 2016-05-27) by slope, elevation and aspect.

Figure 5: Added (left) and (right) in caption after corresponding dates.
Ok

L212: Remove double period.
Ok

L263: "most probably" -> likely
Ok

L308: "modelisation of snow density" -> modeled snow density
Ok

L341: "global datasets only" -> global publicly available atmospheric reanalysis datasets only
We should probably keep "global dataset only" because it is also referring to the Copernicus Land Cover and DEM, as stated in the next sentence.

L349: "near real time" -> near-real-time
Ok

**Bibliography:**
Broxton, P., Ehsani, M. R., & Behrangi, A. (2024). Improving Mountain Snowpack Estimation Using Machine Learning With Sentinel-1, the Airborne Snow Observatory, and University of Arizona Snowpack Data. Earth and Space Science, 11(3), e2023EA002964. https://doi.org/10.1029/2023EA002964

Hoppinen, Z., Palomaki, R. T., Brencher, G., Dunmire, D., Gagliano, E., Marziliano, A., et al. (2024). Evaluating Snow Depth Retrievals from Sentinel-1 Volume Scattering over NASA SnowEx Sites. EGUsphere, 1–35. https://doi.org/10.5194/egusphere-2024-1018

(^ published article should be coming out soon so be on the lookout for that)

Raleigh, M. S., & Small, E. E. (2017). Snowpack density modeling is the primary source of uncertainty when mapping basin-wide SWE with lidar: Uncertainties in SWE Mapping With Lidar. Geophysical Research Letters, 44(8), 3700–3709. https://doi.org/10.1002/2016GL071999